# VQ-Flows: Vector Quantized Local Normalizing Flows

**Sahil Sidheekh**[*1]  **Chris B. Dock**[*2]  **Tushar Jain**[1]  **Radu Balan**[2]  **Maneesh K. Singh**[†3]

[1]Verisk Analytics
[2]University of Maryland, College Park
[3]Motive Technologies, Inc.

## Abstract

Normalizing flows provide an elegant approach to generative modeling that allows for efficient sampling and exact density evaluation of unknown data distributions. However, current techniques have significant limitations in their expressivity when the data distribution is supported on a low-dimensional manifold or has a non-trivial topology. We introduce a novel statistical framework for learning a mixture of *local* normalizing flows as "chart maps" over the data manifold. Our framework augments the expressivity of recent approaches while preserving the signature property of normalizing flows, that they admit exact density evaluation. We learn a suitable atlas of charts for the data manifold via a vector quantized auto-encoder (VQ-AE) and the distributions over them using a conditional flow. We validate experimentally that our probabilistic framework enables existing approaches to better model data distributions over complex manifolds.

## 1 INTRODUCTION

Generative modeling is a machine learning paradigm that aims to learn data distributions and sample from it. If the data is drawn from a random variable $x \sim p(x)$, then one way to do this is to directly model $p(x)$ via a parameterized model ($\theta$) so that $p_\theta(x) \approx p(x)$. Such a model can then be used to generate new samples, which are expected to be statistically indistinguishable from the observed samples. Moreover, generative models that learn $p(x)$ are useful for data augmentation, outlier detection, domain transfer [1, 2], and as priors for other downstream tasks [3, 4, 5].

Among the most successful generative models are deep latent variable models, which assume that the latent factors of variation underlying the generative process of the data follow a simple distribution, such as a Gaussian or a uniform distribution. The non-linear function transforming this latent space to the data space (or vice-versa) is parameterized as a neural network and learned using gradient descent. Depending upon their formulation, there are three broad categories of deep latent variable models - GANs [6], VAEs [7], and normalizing flows.

In this work, we focus on normalizing flows, a class of deep latent variable models introduced in [8] that support efficient sampling, exact density estimation, and inference [9]. A normalizing flow maps the data space to a latent space through a series of diffeomorphisms (differentiable, bijective transformations with differentiable inverses). The data is assumed to follow an analytically computable distribution in the latent space, typically a Gaussian. Since the mapping is a diffeomorphism, the density in the data space can be obtained using the change of variables formula. To generate new samples using a flow, one can sample from the latent distribution and use the inverse transformation to map them to the data space. This makes normalizing flows powerful generative models that support exact density evaluation in contrast to GANs and VAEs.

Despite the advantages of normalizing flows over other generative models, their diffeomorphic requirement poses several restrictions. Firstly, a continuous bijective transformation with continuous inverse preserves the topology of its domain. Therefore, the data space is required to be topologically equivalent to the support of the latent distribution, typically to $D$ dimensional Euclidean space since the latent distribution is assumed to be a Gaussian. However, real data distributions typically differ from Euclidean space in many topological respects, such as the number of connected components, the presence of holes, etc. A normalizing flow would thus fail to model such data distribution accurately. Note, as an aside, that other generative models like GANs also suffer from these topological issues [10].

---

[*]Equal contribution
[†]Work was performed while at Verisk Analytics.

*Accepted for the 38th Conference on Uncertainty in Artificial Intelligence* (UAI 2022).

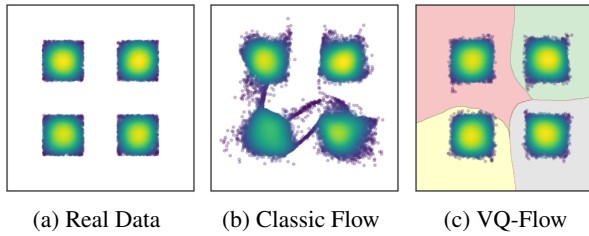

(a) Real Data     (b) Classic Flow     (c) VQ-Flow

Figure 1: Augmentation of our framework (c) enables a classic flow (b) to better model the discontinuities in the data manifold through a learned atlas of charts(shaded region).

A particularly troubling consequence of the continuous invertibility of flow transformations is that they are dimensionality preserving. However, according to the *manifold hypothesis*, high dimensional real-world data living in $\mathcal{X} \simeq \mathbb{R}^D$ is often supported on a $d << D$ manifold of the embedding space. To efficiently learn such distributions using flows, one needs to design expressive transformations that can map from a $d$ dimensional latent space to a the $D$ dimensional data space without making learning intractable. Recent work using stochastically invertible tall matrices [11] and dimension raising conformal embeddings [12] have paved the way in designing such transformations, however in both works expressivity is limited by the fact that the dimension changing operations are restricted to be linear (in [11]) or made up of Möbius transformations (in [12]).

In this work, we propose to address the above limitations by parameterizing a family of normalizing flows to compose an atlas of charts over the data manifold. As the topology of the data manifold is expected to be "locally" equivalent to Euclidean space, a local normalizing flow should be able to model the local distribution over a chart region effectively. Further, by learning a mixture of flows over well-chosen charts, our approach compensates naturally for the limited expressiveness of existing flows. We summarize the main contributions of this work below:

- We provide an understanding of the limited expressive power of existing flow-based models in modeling data distributions over complex topological spaces.

- We present a statistical framework for defining an expressive mixture of local normalizing flows that is flexible and generic enough to be used with existing approaches. We show that this framework allows for efficient sampling, inference of latent variables, and exact density evaluation while improving expressivity.

- We validate experimentally that the proposed approach improves flows for density estimation and sample generation, and is thus able to resolve many of the topological restrictions on expressivity imposed by using global diffeomorphisms.

## 2 BACKGROUND

Given data $\{x_n\}_{n=1}^N \subset \mathcal{X} \simeq \mathbb{R}^D$ distributed according to an unknown distribution $p(x)$, a normalizing flow maps it through a diffeomorphism $f : \mathcal{X} \to \mathcal{Z}$ to a latent space $\mathcal{Z} \simeq \mathbb{R}^D$ such that $z = f(x)$ is simply distributed, for example $z \sim q(z)$ where $q = N(0, \mathbb{I})$. Recall that a diffeomorphism is a differentiable map that is bijective and whose inverse is also differentiable. Typically one denotes by $g$ the inverse of $f$ and parameterizes the normalizing flow as $x = g_\theta(z)$, where $\theta$ is the vector of learnable model parameters. The process of going from the latent space to data space is called *generation* or *sampling* and is accomplished by the function $g_\theta$, while the inverse procedure is termed *inference* and is accomplished by $f_\theta = g_\theta^{-1}$:

$$
\begin{array}{cc}
f_\theta : \mathcal{X} \to \mathcal{Z} & g_\theta : \mathcal{Z} \to \mathcal{X} \\
\underbrace{x \mapsto f_\theta(x)}_{\text{Inference}} & \underbrace{z \mapsto g_\theta(z)}_{\text{Sampling}}
\end{array} \quad (1)
$$

The approximation $p_\theta(x)$ to the true probability density $p(x)$ is then obtained from $q(z)$ through the change of variables formula as:

$$
p_\theta(x) = q(f_\theta(x))|\det[Jf_\theta(x)]| \quad (2)
$$

As compositions of diffeomorphisms are also diffeomorphisms, one can design expressive flows by composing individual transformations that have simple to compute inverses and Jacobian determinants. Suppressing the vector of model parameters $\theta$, we will use the notation $f(x) = f^1 \circ \cdots \circ f^L(x)$ where $f^1, \ldots, f^L$ are assumed to have easily computable Jacobian determinants and inverses. Define recursively $x^{l-1} = f^l(x^l)$, $1 \le l \le L$, with $x^L = x$. Note that $x^l = f^{l+1} \circ \cdots \circ f^L(x)$ and $x^0 = f(x)$. One can then write the log-likelihood as:

$$
\log p(x) = \log q(z) + \log \prod_{l=1}^L |\det[Jf^l(x^l)]| 
$$
$$
= \log q(f(x)) + \sum_{l=1}^L \log |\det[Jf^l(x^l)]| \quad (3)
$$

A given layer $f^l$ of the normalizing flow will depend only on a subset $\theta_l$ of the parameters of $\theta := (\theta_1, \ldots, \theta_L)$. Temporarily adding back in the $\theta$ dependence of $f_\theta$, maximum likelihood estimation of $\theta$ then yields the following optimization problem:

$$
\theta^* = \min_{\theta=(\theta_1,\ldots,\theta_L)} \frac{1}{N} \sum_{n=1}^N - \log p_\theta(x_n)
$$
$$
= \min_{\theta=(\theta_1,\ldots,\theta_L)} \frac{1}{N} \sum_{n=1}^N \Big\{ - \log q(f_\theta(x_n)) \quad (4)
$$
$$
- \sum_{l=1}^L \log |\det[Jf_{\theta_l}^l(x_n^l)]| \Big\}
$$

# 3 RELATED WORK

Normalizing flows have come a long way since it was introduced in [9, 13], with much efforts focused on expanding their scalability and applicability. This has resulted in several different formulations [14, 15, 16, 17], each with a multitude of proposed architectures [18, 19, 20, 21, 22, 23], aimed at defining expressive yet analytically invertible flow transformations with efficiently computable jacobian determinants. However, as these approaches define invertible transformations in Euclidean space, they are dimensionality preserving and less suited for modeling distributions over lower dimensional manifolds [24, 25]. Subsequent works have tried to address this challenge by building injective flows [26, 11, 27, 28, 29, 30]. However, they trade off the benefits of dimensionality change to intractable density estimation or stochastic inverses. The work by [12] overcomes the above limitations using conformal embeddings, but has limited expressive power, as we show in this work. One way to improve the expressivity of all the above approaches, and enable them to overcome topological constraints [31], is to relax their global diffeomorphic requirement by defining a *mixture of flows*. Prior works in this direction have looked at infinite mixtures by defining flows in a lifted space [32] or by using continuous indexing [33]. However, their added expressivity comes at the cost of tractable density computation, and one has to rely on variational approximations to train the model. A manifold geometric approach to normalizing flows is also taken in [34] and [35], however in contrast to this work these techniques assume the manifold and its Riemannian geometry are known. On similar lines with this work, [36] proposes to use a finite mixture of flows through piecewise-invertible transformations over partitions of the data space by introducing both real and discrete valued latent variables in the flow. However, this formulation introduces discontinuities in the model density that leads to unstable training [33], necessitating the enforcement of boundary conditions through ad-hoc architectural changes. It is therefore limited in its generalizability to novel flow formulations. Our work, on the other hand, by decoupling the partition learning from the flow training, introduces a more generic and scalable framework that can aid existing flows to overcome topological constraints and learn complex data distributions efficiently.

# 4 METHODOLOGY

A traditional normalizing flow provides a global diffeomorphism between the latent space $\mathcal{Z}$ and the data space $\mathcal{X} \simeq \mathbb{R}^D$, and as such requires the latent space to have the same dimension as the data space. This can lead to numerical instability when the data is supported on a $d < D$ dimensional manifold $\mathcal{M} \subset \mathcal{X}$ because the learned transformation will tend to become "less and less injective" as it seeks to restrict its range to $\mathcal{M}$ [24, 25].

One way to overcome this challenge is to build transformations that map across dimensions while preserving invertibility on its image. Unfortunately, the natural approach of post-composing a $d$ dimensional bijective normalizing flow $g : \mathcal{Z} \to \mathcal{U}$ with a dimension-raising embedding $e : \mathcal{U} \to \mathcal{X}$ leads in general to an intractable likelihood since the determinant in the change of variables formula $p(x) = q(f(x))|Det[J_g J_e^T J_e J_g]|^{-\frac{1}{2}}$ no longer separates into a product of simpler determinants. We will focus on the solution to this issue developed in [12], namely to post-compose the $d$ dimensional bijective normalizing flow $g : \mathcal{Z} \to \mathcal{U}$ with a dimension raising *conformal embedding* $c : \mathcal{U} \to \mathcal{X}$. An alternative solution developed in [11] is to use a linear dimension raising embedding and invert it stochastically, but this approach relies on the dimension change operation being linear which is restrictive. The approach taken in [12] hinges on the fact that for every $u \in \mathcal{U}$ the Jacobian $J_c(u)$ satisfies $J_c(u)^T J_c(u) = \lambda(u)^2 \mathbb{I}$ for $\lambda : \mathcal{U} \to \mathbb{R}$, thus

$$\begin{aligned}
\det[J_{c \circ g}^T J_{c \circ g}]^{\frac{1}{2}} &= \det[J_g^T J_c^T J_c J_g]^{\frac{1}{2}} \\
&= |\lambda(u)| \det[J_g^T J_g]^{\frac{1}{2}} \\
&= |\lambda(u)| |\det[J_g]|
\end{aligned} \tag{5}$$

This splitting keeps the likelihood computation tractable, but the requirement that $\mathcal{M}$ be the range of a conformal embedding is artificially restrictive. This issue is exacerbated by the necessity of parameterizing $c$. As noted in [12] the easiest way to do so is to let $c = c_J \circ \cdots \circ c_1$ where each $c_j$ is either a trivially conformal zero padding operation or a dimension preserving conformal transformation. A dimension preserving conformal transformation $f : \mathbb{R}^d \to \mathbb{R}^d$ with $d > 2$ is restricted by Liouville's theorem to be a Möbius transformation, of the form $f(x) = (A, a, b, \alpha, \epsilon)(x) = b + \alpha(Ax - a)/||Ax - a||^\epsilon$ where $A \in O(d)$ is an orthogonal matrix, $\alpha \in \mathbb{R}$, $a, b \in \mathbb{R}^d$, and $\epsilon$ is either 0 or 2. Though it might initially appear that the composition of many such operations would give increased expressive power, the group structure of the Möbius transformations prevents this. Indeed, if $p_s : \mathbb{R}^d \to \mathbb{R}^{d+s}$ is the zero padding operation, $m_1 = (A_1, a_1, b_1, \alpha_1, \epsilon_1)$ is a $d$ dimensional Möbius transformation and $m_2 = (A_2, a_2, b_2, \alpha_2, \epsilon_2)$ is a $d+s$ dimensional Möbius transformation then it is easily verified that for $x \in \mathbb{R}^d$

$$m_2 \circ p_s \circ m_1(x) = (m_2 \cdot \tilde{m}_1)(p_s(x)) \tag{6}$$

Where $\tilde{m}_1$ is the $d + s$ dimensional Möbius transformation

$$\tilde{m}_1 = \left( \begin{bmatrix} A_1 & 0 \\ 0 & \mathbb{I}_{s \times s} \end{bmatrix}, p_s(a_1), p_s(b_1), \alpha_1, \epsilon_1 \right) \tag{7}$$

Thus, this parametrization yields $c$ as a Möbius transformation of $\mathbb{R}^D$ composed with $p_{D-d}$. Practically speaking, if $c$ is parameterized as above, the assumption that $\mathcal{M}$ is the image of a global conformal embedding severely limits expressiveness. The class of global conformal embeddings is

not subject to Liouville's theorem and is far richer than the set of Möbius transformations, but it is hard to parameterize.

## 4.1 DIFFERENTIAL GEOMETRY OF CONFORMALLY FLAT MANIFOLDS

A weaker and more natural assumption than $\mathcal{M}$ being the image of a conformal embedding is that $\mathcal{M}$ is *locally conformally flat*. Recall that if $f : (\mathcal{N}, \eta_1) \to (\mathcal{M}, \eta_2)$ is a map between differentiable manifolds $\mathcal{N}$ and $\mathcal{M}$ with metrics $\eta_1 : \mathcal{N} \times T\mathcal{N} \times T\mathcal{N}$ and $\eta_2 : \mathcal{M} \times T\mathcal{M} \times T\mathcal{M}$ respectively then the pullback $f^*\eta_2$ of the metric $\eta_2$ through $f$ is defined via:

$$f^*\eta_2 : \mathcal{N} \times T\mathcal{N} \times T\mathcal{N} \to \mathbb{R}$$
$$f^*\eta_2(y, v, w) = \eta_2(f(y), Df(y)(v), Df(y)(w)) \tag{8}$$

With this in mind a $d$ dimensional manifold $\mathcal{M}$ is called *locally conformally flat* if $\eta_1 = \sum_{i=1}^{d} dy_i^2$ is the flat metric and for any $x \in \mathcal{M}$ there is a neighborhood $U \ni x$, an open set $O \subset \mathbb{R}^d$, a diffeomorphism $f : O \to U$, and a differentiable scalar function $\lambda : O \to \mathbb{R}$ such that $f^*\eta_2(y, \cdot, \cdot) = \lambda(y)\eta_1(\cdot, \cdot)$ for all $y \in O$ [37]. An alternate definition replaces $\mathbb{R}^d$ with a flat manifold (defined as having an identically vanishing Riemannian curvature tensor), but this definition is equivalent to the above since any $d$ dimensional flat manifold is locally isometric to $\mathbb{R}^d$ (not globally isometric, for example tori are flat when equipped with appropriate coordinates) [38]. In our case the metric $\eta_2$ is assumed to be inherited from the Euclidean metric on $\mathcal{X} \simeq \mathbb{R}^D$.

The notion of local conformal flatness provides far more flexibility than its global counterpart. It is well known, for example, that every 2 dimensional Riemannian manifold is locally conformally flat, but even the sphere $S^2(\mathbb{R})$ is not globally conformally flat (by contrast an explicit local conformal equivalence of $S^d(\mathbb{R})$ to $\mathbb{R}^d$ is given by stereographic projection from the north and south poles) [38]. In general, criteria are known for a Riemannian manifold of dimension $d > 2$ to be locally conformally flat: For $d = 3$ a pseudo-Riemanian manifold is locally conformally flat if and only if the Cotton tensor vanishes everywhere, for $d \geq 4$ a pseudo-Riemannian manifold is locally conformally flat if and only if the Weyl tensor vanishes everywhere [38]. The question of which manifolds are globally conformally flat is more difficult, and in applied problems this requirement is artificially restrictive.

## 4.2 LOCAL NORMALIZING FLOWS

We propose to break up the data manifold $\mathcal{M}$ into an atlas of overlapping charts $V_1, \ldots, V_K$.

**Definition 1** (See [37]). *An atlas of (smooth) charts for a $d$ dimensional manifold $\mathcal{M}$ is a collection of subsets of $\mathcal{M}$,* $\{V_k\}_{k=1}^K$ *and a collection of subsets* $\{P_k\}_{k=1}^K$ *of* $\mathbb{R}^d$ *such that* $\bigcup_{k=1}^K V_k = \mathcal{M}$ *and a collection of invertible maps* $f_k : V_k \to P_k$ *such that the "transition maps"* $f_i \circ f_j^{-1} :$ $f_j(V_i \cap V_j) \to f_i(V_i \cap V_j)$ *are smooth.*

We will assume charts of the form $V_j = U_j \cap \mathcal{M}$ and $P_j = f_j(V_j)$ where $U_j$ are learned open subsets of $\mathcal{X}$ such that $\{x_n\}_{n=1}^N \subset \bigcup_{k=1}^K U_k$ and $f_1, \ldots, f_K$ are conformal normalizing flows. In a slight abuse of terminology we will also refer to $U_1, \ldots, U_k$ as charts. To handle dimensionality change, we assume that the manifold $\mathcal{M}$ is locally conformally flat and of dimension $d$, implying that for $V_j$ sufficiently small there exists $D_k \subset \mathcal{U}$ and a conformal dimension raising map $c_k : \mathcal{U} \to \mathcal{X}$ so that $V_k = c_k(D_k) = c_k \circ g_k^L \circ \cdots \circ g_k^1(P_k)$.

Because chart regions may in general overlap, we propose to choose between them probabilistically. Specifically, given $U_1, \ldots, U_K$ that cover the data manifold $\mathcal{M}$, we model $p(x)$ via a latent random variable $z$ that takes values in $\mathcal{Z}$ and a "chart picking" discrete random variable $k$ that takes values in $\{1, \ldots, K\}$. For $k = 1, \ldots, K$ let $g_k : \mathcal{Z} \to U_k$ be a global immersion (a differentiable injection whose Jacobian is everywhere full rank) with left inverse $f_k : V_k \to \mathcal{Z}$ and range $V_k = g_k(P_k) = \mathcal{M} \cap U_k$.

**Proposition 1.** *Let* $(U_k)_{k=1}^K$, $(V_k)_{k=1}^K$, $(g_k)_{k=1}^K$, *and* $(f_k)_{k=1}^K$ *be as above. Further, let $k$ be a discrete random variable taking values $1, \ldots, K$ and $z$ a continuous random variable taking values in $\mathcal{Z}$. Then if $x$ is a random variable supported on $\mathcal{M}$ such that*

$$p(x, z, k) = \delta(x - g_k(z))q(z)p_k \tag{9}$$

*One has*

*(i) The joint distribution of $x$ and $k$ is given by:*

$$p(x, k) = p_k \mathbb{1}_{V_k}(x) |\det[Jf_k(x)Jf_k(x)^T]|^{\frac{1}{2}} q(f_k(x)) \tag{10}$$

*(ii) The marginals $p(k)$ and $p(z)$ are given by $p_k$ and $q(z)$ respectively.*

*(iii) The random variables $z$ and $k$ are independent.*

*(iv) The conditional distributions $p(x|k)$ and $p(k|x)$ are given by:*

$$p(x|k) = \mathbb{1}_{V_k}(x) |\det[Jf_k(x)Jf_k(x)^T]|^{\frac{1}{2}} q(f_k(x)) \tag{11}$$

$$p(k|x) = \frac{p_k \mathbb{1}_{V_k}(x) |\det[Jf_k(x)Jf_k(x)^T]|^{\frac{1}{2}} q(f_k(x))}{\sum_{j:x \in V_j} p_j |\det[Jf_j(x)Jf_j(x)^T]|^{\frac{1}{2}} q(f_j(x))} \tag{12}$$

*(v) The density of interest, $p(x)$ is given by*

$$p(x) = \sum_{k:x \in V_k} p_k |\det[Jf_k(x)Jf_k(x)^T]|^{\frac{1}{2}} q(f_k(x)) \tag{13}$$

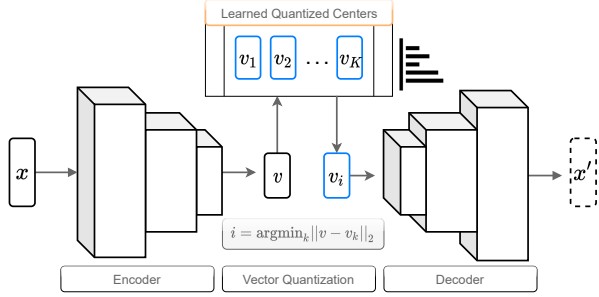

Figure 2: Learning quantized centers on the low dimensional data manifold using a vector quantized auto-encoder.

*Proof.* Deferred to the appendix (see Section 7). □

Thus we assume the joint distribution of $x$, $z$, and $k$ to be $p(x, z, k) = \delta(x - g_k(z))q(z)p_k$ and apply the above proposition. We will use either $q = N(0, \mathbb{I})$ or $q = \frac{1}{vol(B_1(0))}\mathbb{1}_{B_1(0)}$ as our latent distribution and let $p_k$ be the normalized probability with which $x$ occurs in $U_k$, that is:

$$p_k := \frac{p(x \in U_k)}{\sum_{j=1}^{K} p(x \in U_j)} = \frac{\int_{U_k} p(x)dx}{\sum_{j=1}^{K} \int_{U_j} p(x)dx} \quad (14)$$

It remains to learn a "good" collection of charts $U_1, \ldots, U_K$, estimate $p_1, \ldots, p_K$, and then to parameterize $g_1, \ldots, g_K$ via normalizing flows $g_1^\theta, \ldots, g_K^\theta$ and obtain a maximum likelihood estimate for $\theta$ by optimizing $-\log p_\theta(x)$ (where $p_\theta(x)$ is as in (13)).

#### 4.2.1 Learning the collection of charts $U_1, \ldots, U_K$

We learn the charts $U_1, \ldots, U_K$ via a vector-quantized auto encoder (VQ-AE)[39], as it provides an effective and scalable mechanism to learn quantized centers on lower dimensional manifolds (also see [40] for a recent application on high-dimensional data). The VQ-AE learns an encoder map $E : \mathcal{X} \to \mathcal{V}$, a decoder map $D : \mathcal{V} \to \mathcal{X}$, and a collection of "encoded chart centers" $Q = \{v_k\}_{k=1}^{K} \subset \mathcal{V}$ that minimize the reconstruction error $\mathcal{L}(D(\arg\min_{v \in Q} ||v - E(x)||_2), x)$. Once $D$, $E$, and $Q$ are learned we compute $d_k(x) = ||E(x) - v_k||_2$ for $k = 1, \ldots K$. With $d_1(x), \ldots d_k(x)$ in hand it remains to compute our charts. We would like the charts to overlap, but we also want them to be sparse in the sense that no individual $x$ has too many relevant charts. One possible choice is to fix $m \in \{1, \ldots, K\}$ and let $\tilde{d}_1 \leq \cdots \leq \tilde{d}_K$ be the sorted permutation of $d_1, \ldots, d_K$ then define $U_k = \{x : ||E(x) - v_k||_2 \leq \tilde{d}_m(x)\}$, so that every point $x$ has at least $m$ charts associated to it (those whose encoded chart centers are among the $m$ closest to $E(x)$). With this choice, a point $x$ will have exactly $m$ associated charts so long as the $m^{th}$ closest chart center is unique. Another choice would be to fix $\epsilon > 0$ and let

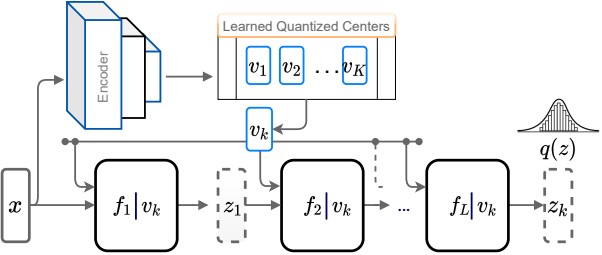

Figure 3: Learning the data distribution using a family of normalizing flows conditioned on the quantized centers.

$U_k = \{x : ||E(X) - v_k||_2 < (1 + \epsilon)\tilde{d}_m(x)\}$ (increasing $\epsilon$ enlarges each chart). For now we leave $m$ and $\epsilon$ as hyperparameters, and in general denote $m(x) = |\{k : x \in U_k\}|$ (one always has $m(x) \geq m$). Note that checking if $x \in U_k$ amounts to computing $E(x)$ and $\tilde{d}_1(x), \ldots, \tilde{d}_K(x)$ and verifying that $||E(x) - v_k||_2 < (1 + \epsilon)\tilde{d}_m(x)$.

#### 4.2.2 Estimating $p_1, \ldots, p_K$

Once $U_1, \ldots, U_K$ are fixed note that if $r_k := p(x \in U_k)$,

$$r_k = \mathbb{E}_{x \sim p(x)}[\mathbb{1}_{U_k}(x)] \quad (15)$$

The density $p(x)$ is unknown at this point, but we may estimate $r_k$ using the empirical distribution $\rho(x) = \frac{1}{N}\sum_{n=1}^{N} \delta(x - x_n)$ so that $r_k \approx \mathbb{E}_{x \sim \rho(x)}[\mathbb{1}_{U_k}(x)]$. Practically speaking we thus perform a second pass over the training data and update $r_1, \ldots, r_K$ (initialized as zero) via $r_k^{(n)} = \frac{n-1}{n}r_k^{(n-1)} + \frac{1}{n}\mathbb{1}_{U_k}(x_n), 1 \leq n \leq N$, finally setting $r_k = r_k^{(N)}$ and $p_k = r_k / \sum_{j=1}^{K} r_j$.

#### 4.2.3 Learning the local transformations $g_1, \ldots, g_K$

Once $U_1, \ldots, U_K$ and $p_1, \ldots, p_K$ are obtained we model $g_k : \mathcal{Z} \to U_k$ as an $L$ layered invertible conditional normalizing flow. Where dimensionality change is required, we post-compose it with a conformal dimension raising map so that $g_k = c_k \circ g_k^L \circ \cdots \circ g_k^1$. We write the left inverse of $g_k$ via $f_k = f_k^1 \circ \cdots \circ f_k^L \circ c_k^\dagger$ where $f_k^l = (g_k^l)^{-1}$ and $c_k^\dagger$ denotes the left inverse of the conformal map $c$ obtained by removing the zero padding and inverting the various Möbius transformations composing $c_k$. In practice, we reduce the number of parameters of our model by restricting each $g_k^l$ (and $f_k^l$) to depend on $k$ only through the value of the encoded chart center $v_k$. With this parametrization of $f_1, \ldots, f_K$ in hand (11) becomes

$$p(x|k) = \mathbb{1}_{V_k}(x)q(f_k(x))|\lambda_k(c_k^\dagger(x))|^{-1}$$
$$\prod_{l=1}^{L} |\det[Jf_k^l(f_k^{l+1} \circ \cdots \circ f_k^L(x))]| \quad (16)$$

where $\lambda_k(u)$ is defined via $(Jc_k(u))^T(Jc_k(u)) = \lambda_k(u)^2\mathbb{I}$.

As we'll see this approach allows for far higher expressive power than global conformal flows without sacrificing the ability to generate realistic samples, perform inference, or compute exact densities. Indeed we may rewrite (13) via

$$p(x) = \sum_{k:x \in U_k} p(x|k)p(k)$$
$$= \mathbb{E}_{k \sim \tilde{p}_x(k)}[p(x|k)] \underbrace{\sum_{j:x \in U_j} p(j)}_{\text{piecewise constant}} \quad (17)$$

Where $\tilde{p}_x(k) = p(k|p(x|k) > 0) = p(k)/\sum_{j:x \in U_j} p(j)$. Thus, during *training* of the conditional normalizing flow we may replace the expectation $\mathbb{E}_{k \sim \tilde{p}(k)}[p(x|k)]$ with the stochastic quantity $p(x|k), k \sim \tilde{p}(k)$, performing only a single gradient descent pass per data-point as opposed to $m(x)$ passes. If the exact likelihood is needed, however, it can be computed at the cost of evaluating the normalizing flow and its Jacobian $m(x)$ times:

$$p(x) = \sum_{k:x \in U_k} p(x|k)p(k)$$
$$= \sum_{k:x \in U_k} p_k q(f_k(x))|\lambda_k(c_k^\dagger(x))|^{-1} \quad (18)$$
$$\prod_{l=1}^{L} |\det[Jf_k^l(f_k^{l+1} \circ \cdots \circ f_k^L(x))]|$$

Since $z$ and $k$ are independent, one can perform the *sampling task* via first sampling $z \sim q(z)$ and $k \sim p(k)$ and then computing a single forward pass of the normalizing flow chosen by $k$ to obtain $x = g_k(z)$.

The *inference task* is complicated slightly by the fact that $z$ is no longer wholly determined given $x$, but instead takes values $(f_k(x))_{k:x \in U_k}$ with corresponding probabilities $(p(k|x))_{k:x \in U_k}$. One could perform a stochastic inference via sampling $k \sim p(k|x)$ and computing $z = f_k(x)$ (this amounts to choosing among the relevant charts for $x$), however if deterministic inference is preferred then of course one may always compute the expected value of $z$ as $z = \mathbb{E}_{k \sim p(k|x)}[f_k(x)] = \sum_{k:x \in U_k} p(k|x)f_k(x)$ or the most probable value of $z$ as $z = f_s(x)$ where $s = \text{argmax}_{k:x \in U_k} p(k|x)$.

### 4.3 HARD-BOUNDARY OR DETERMINISTIC APPROXIMATION

A particularly simple special case of the above model is the case $m = 1$ and $\epsilon = 0$, in which only a single chart is associated to a given $x$. This case reduces our atlas of overlapping charts to a disjoint partition of the data manifold $\mathcal{M}$. In this case $U_k$ is exactly the subset of $\mathcal{X}$ for whom $E(x)$ is closest to the encoded chart center $v_k$, and thus with the exception of $x$ lying on the chart

boundaries, the random variable $k$ can be treated as a deterministic function of the random variable $x$, namely $k(x) = \text{argmin}_{k=1,...,K} ||E(x) - v_k||_2 = \sum_{k=1}^{K} k \mathbb{1}_{U_k}(x)$. Sampling in the hard-boundary case is identical to sampling in the soft-boundary case: generate samples for $x$ by first sampling $z \sim q(z)$ and $k \sim p(k)$ and then computing $x = g_k(z)$. Inference in the hard-boundary case is unambiguous since

$$\mathbb{E}_{k \sim p(k|x)}[f_k(x)] = f_s(x)$$
$$s = \underset{k=1,...,K}{\text{argmax}} \, p(k|x) = \underset{k=1,...,K}{\text{argmin}} \, ||E(x) - v_k||_2 \quad (19)$$

That is to say that one performs inference by first identifying which region $R_s$ contains $x$ and then computing $z = f_s(x)$. The most significant simplification in the hard-boundary case from a computational standpoint comes in computing the likelihood $p(x)$, since if $x \in U_k$ then

$$p(x) = p(x,k) = p(x|k)p(k)$$
$$= p(k)q(f_k(x))|\lambda_k(c_k^\dagger(x))|^{-1}$$
$$\prod_{l=1}^{L} |\det[Jf_k^l(f_k^{l+1} \circ \cdots \circ f_k^L(x))]| \quad (20)$$

Thus only one normalizing flow needs to be evaluated to compute the exact likelihood $p(x)$ (as opposed to $m(x)$ of them) and the normalizing flows may be trained using the exact likelihood as opposed to an unbiased estimator for it.

## 5 EXPERIMENTS

To experimentally validate the efficacy of the proposed framework, we consider six 3-dimensional data distributions over manifolds of varying complexity as shown in Figure 4. Each dataset consists of $10,000$ datapoints, $5,000$ of which we use for training and $2,500$ each for validation and testing. We train three different normalizing flows - RealNVP [13], Masked Autoregressive Flows (MAF) [18] and Conformal Embedding Flows (CEF) [12] over these datasets with and without the augmentation of our framework. We refer to a base *flow* augmented with the vector quantized conditioning as VQ-*flow*. We define each model using 5 flow transformations and train them for 100 epochs using an Adam optimizer, early stopping if the validation performance does not improve over 10 epochs. For CEF, we use a 2-dimensional RealNVP as the base flow, which is then raised to the 3-dimensional space using the conformal embedding. We parameterize the VQ-AE using feedforward neural networks and use a latent dimension of 2 with $k = 32$, to learn the partitioning of the data manifold. To define the conditional normalizing flow, we use the parameterization given in [41]. We evaluate the models for density estimation and sample generation. We follow the same hyperparameters for a base flow and its VQ-counterpart without any tuning and report the performance averaged over 5 independent trials. We defer further details on data generation,

| Model | Spherical | Helix | Lissajous | Twisted-Eight | Knotted | Interlocked-Circles |
|---|---|---|---|---|---|---|
| Real NVP | $3.15 \pm 0.07$ | $-3.37 \pm 0.16$ | $2.42 \pm 0.07$ | $0.94 \pm 0.15$ | $-2.17 \pm 0.14$ | $0.95 \pm 0.13$ |
| VQ-RealNVP | $3.55 \pm 0.04$ | $-1.66 \pm 0.08$ | $3.04 \pm 0.15$ | $2.29 \pm 0.14$ | $0.39 \pm 0.18$ | $2.42 \pm 0.25$ |
| MAF | $4.38 \pm 0.10$ | $-2.90 \pm 0.02$ | $2.50 \pm 0.12$ | $1.34 \pm 0.22$ | $-1.02 \pm 0.14$ | $1.07 \pm 0.07$ |
| VQ-MAF | $4.43 \pm 0.14$ | $-0.49 \pm 0.03$ | $3.48 \pm 0.16$ | $2.01 \pm 0.10$ | $0.62 \pm 0.16$ | $2.29 \pm 0.18$ |
| CEF | $0.91 \pm 0.07$ | $-3.71 \pm 0.09$ | $0.42 \pm 0.15$ | $-0.38 \pm 0.21$ | $-2.48 \pm 0.26$ | $-0.72 \pm 0.11$ |
| VQ-CEF | $0.98 \pm 0.11$ | $-2.90 \pm 0.17$ | $1.65 \pm 0.14$ | $-0.32 \pm 0.19$ | $-1.93 \pm 0.17$ | $1.24 \pm 0.15$ |

Table 1: Quantitative evaluation of **Density Estimation** in terms of the test log-likelihood in nats (higher the better) on the 3D datasets. The values are averaged across 5 independent trials, $\pm$ represents the 95% confidence interval.

| Model | Spherical | Helix | Lissajous | Twisted-Eight | Knotted | Interlocked-Circles |
|---|---|---|---|---|---|---|
| Real NVP | $0.50 \pm 0.07$ | $-57.46 \pm 2.11$ | $0.18 \pm 0.14$ | $-2.72 \pm 0.90$ | $-8.65 \pm 0.87$ | $-2.18 \pm 0.37$ |
| VQ-RealNVP | $0.99 \pm 0.14$ | $-3.85 \pm 0.98$ | $0.59 \pm 0.08$ | $0.18 \pm 0.17$ | $-1.44 \pm 0.37$ | $-0.11 \pm 0.12$ |
| MAF | $0.65 \pm 0.26$ | $-92.83 \pm 5.69$ | $0.12 \pm 0.16$ | $-2.77 \pm 0.81$ | $-7.04 \pm 0.49$ | $-2.49 \pm 0.14$ |
| VQ-MAF | $1.01 \pm 0.07$ | $-4.62 \pm 0.37$ | $0.59 \pm 0.07$ | $-0.32\pm 0.13$ | $-2.44 \pm 0.11$ | $-0.15 \pm 0.08$ |
| CEF | $-1.17 \pm 0.06$ | $-29.90 \pm 2.12$ | $0.38 \pm 0.14$ | $-4.03 \pm 0.38$ | $-19.40 \pm 1.80$ | $-3.42 \pm 0.49$ |
| VQ-CEF | $0.80 \pm 3.42$ | $-20.75 \pm 2.22$ | $0.49 \pm 0.03$ | $-3.51 \pm 0.73$ | $-14.44 \pm 1.57$ | $-3.23 \pm 0.19$ |

Table 2: Quantitative evaluation of **Sample Generation** in terms of the log-likelihood of generated samples in nats (higher the better) on the 3D datasets. The values are averaged across 5 independent trials, $\pm$ represents the 95% confidence interval.

implementation as well as results on additional 3D data distributions to the supplementary material.

## 5.1 DENSITY ESTIMATION

The ability to compute exact likelihood is one of the critical features of a normalizing flow that makes it a potential tool in solving inverse problems. Improving the expressive power of flows can thus enhance their utility as priors by better modeling the data density. Thus, we first evaluate the proposed framework's ability to enhance the expressivity of flows to perform better density estimation. Table 1 compares the log-likelihood (in nats) achieved by different flow models with and without the VQ-augmentation on a held-out test set. A higher value indicates a better learned density. We observe that VQ-flows are able to achieve higher test log-likelihoods than their non-VQ-counterparts consistently across the considered data distributions. Thus, our framework enables better density estimation for normalizing flows over complex manifolds.

## 5.2 SAMPLE GENERATION

A key desiderata of an expressive generative model is its ability to generate high fidelity samples from the data distribution. Figure 4 visualizes the samples generated by a RealNVP flow trained on the 3D data distributions with and without the VQ augmentation. We observe that while the classical flow is able to generate samples from the data man-

ifold, it also generates data points off the manifold, resulting in a poor fit to the real data distribution, as expected due to the requirements of being a global diffeomorphism. VQ-flows are seen to overcome these restrictions and generate samples better approximating the real data distribution. For a quantitative comparison, we evaluate the log-likelihood of the generated samples using a kernel density estimator fitted on the training data. We use a Gaussian kernel, with an optimal bandwidth obtained through cross-validation for each data distribution. We observe (Table 2) that VQ-flows, owing to their ability to model the topology of the data manifold better, significantly outperform their non-VQ counterparts on sample generation.

## 5.3 HIGH DIMENSIONAL DATA

To study the scalability of the proposed approach to higher dimensions, we consider the MNIST [42] dataset comprising 60,000 grayscale images of handwritten digits, each of dimension 784 ($28\times28$). We train RealNVP and MAF with and without the VQ-augmentation and plot FID scores of the generated samples across their training iterations in Figure 5. We observe that VQ-flows are able to achieve better performance (lower FID scores) faster than their non-VQ counterparts, hence validating the utility of the proposed approach in higher dimensions. An interesting observation here is that while MAF results on MNIST are much better than that of RealNVP, both VQ-MAF and VQ-RealNVP converge to the similar (low) FID scores. This early result

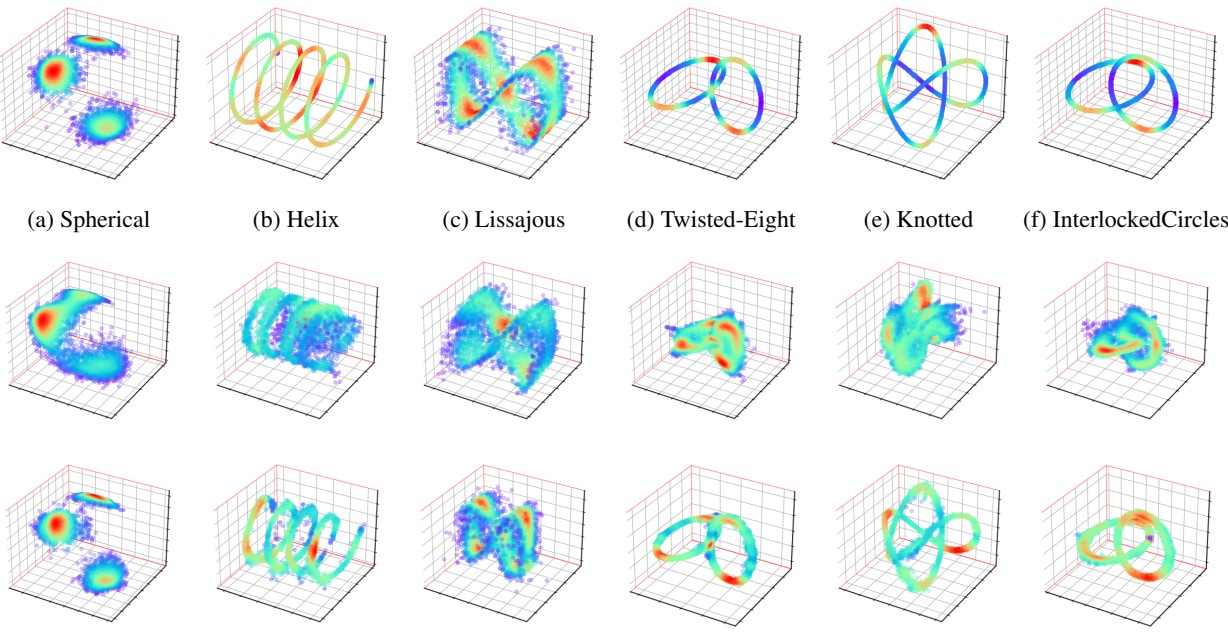

(a) Spherical     (b) Helix     (c) Lissajous     (d) Twisted-Eight     (e) Knotted     (f) InterlockedCircles

Figure 4: Qualitative visualization of the samples generated by a classical flow - RealNVP (Middle Row) and its VQ-counterpart (Bottom Row) trained on Toy 3D data distributions (Top Row).

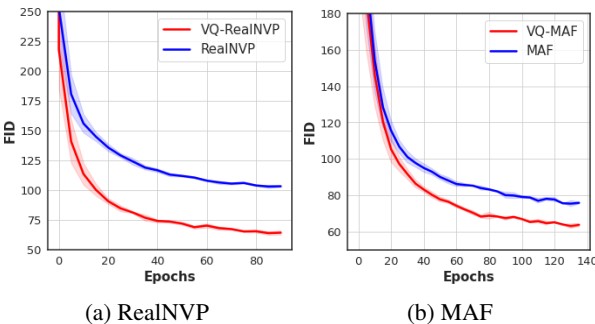

(a) RealNVP           (b) MAF

Figure 5: FID scores (lower the better) across the training of (a) RealNVP and (b) MAF on the MNIST dataset. The shaded region represents the standard deviation over 3 trials.

seems to validate our hypothesis that the core difficulties (topology, dimensionality, etc), even on real datasets, can perhaps be better addressed by the proposed research direction than by improving backbone 'single' flow models.

### 5.4 ABLATION STUDY

Parameterizing the partitioning function using a VQ-AE is a design choice and the no. of partitions $k$ to consider over the data manifold is an important hyperparameter underlying the proposed framework. We conduct an ablation study to evaluate the sensitivity of our approach on $k$ and the partitioning method. We consider k-means clustering

as an alternative design choice for the partitioning function. We train a RealNVP flow over the HELIX data distribution using k-means and VQ-AE, across increasing values of $k$. We plot the validation log-likelihood post training for 25 epochs as a function of $k$ in Figure 6. We observe that VQ-AE results in better performance of the flow consistently across $k$, over k-means. Further, the choice of $k$ beyond a threshold does not have any significant effect on the model, hence it is sufficient to fix it to a large enough value.

## 6 FUTURE WORK & CONCLUSION

Our framework is particularly well suited to high dimensional datasets (such as natural images) that obey the manifold hypothesis, an avenue we hope to explore in the sequel. One of the practical issues we encountered with our approach is that training $g_k$ only on samples from $U_k$ does not always restrict the learned $p(x|k)$ to be supported only on $U_k$. In such cases, the sum over $k$ such that $x \in U_k$ in (18) yields an underestimate for $p(x)$, and the total sum $k = 1, \ldots, K$ must be used instead during testing. In the future, we hope to address this issue by explicitly discouraging the generation of samples outside $U_k$.

To summarize, motivated by differential and conformal geometry, we have developed a novel probabilistic framework for "local" flows. We have demonstrated experimentally on toy data distributions with various topological features that this framework outperforms global flows - both dimen-

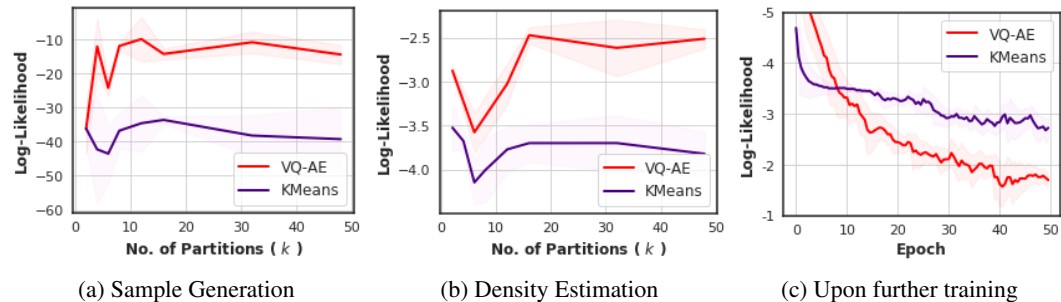

(a) Sample Generation          (b) Density Estimation          (c) Upon further training

Figure 6: **Ablation Study** on the effect of the partitioning method and the number of partitions $k$ on sample generation (a) and density estimation (b). (c)-The learning trajectory of the flow for a fixed $k(=32)$, in terms of validation log-likelihood. The shaded region represents the standard deviation over 3 independent trials.

sion preserving (bijective flows) and dimension raising (embedding flows). Our framework is agnostic to the type of flow transformation employed and retains the key feature of normalizing flows: exact density evaluation. As such, we argue that using local flows as probabilistic chart maps over the data manifold is a natural way to overcome limited expressivity in the presence of dimension change or other topological impediments.

# 7   APPENDIX

*Proof of Proposition 1.* Proving $(i)$.

We can compute the joint distribution over $x$ and $k$ - $p(x, k)$ as given below:

$$p(x, k) = \int_{\mathcal{Z}} p(x, z, k) dz = p_k \int_{\mathcal{Z}} \delta(x - g_k(z)) q(z) dz$$
$$= p_k \mathbb{1}_{V_k}(x) \times$$
$$\int_{\mathcal{Z}} \delta(z - f_k(x)) |\det[Jg_k(z)^T Jg_k(z)]|^{-\frac{1}{2}} q(z) dz$$
$$= p_k \mathbb{1}_{V_k}(x) |\det[Jg_k(f_k(x))^T Jg_k(f_k(x))]|^{-\frac{1}{2}} q(f_k(x))$$
$$= p_k \mathbb{1}_{V_k}(x) |\det[Jf_k(x) Jf_k(x)^T]|^{\frac{1}{2}} q(f_k(x)) \quad (21)$$

Proving $(ii)$. It is readily verified that $p(z) = q(z)$ and $p(k) = p_k$, in particular:

$$p(z) = \sum_{k=1}^{K} \int_{\mathcal{X}} p(x, z, k) dx$$
$$= \sum_{k=1}^{K} p_k \int_{\mathcal{X}} \delta(x - g_k(z)) q(z) dx \quad (22)$$
$$= q(z) \sum_{k=1}^{K} p_k = q(z)$$

and,

$$p(k) = \int_{X} p(x, k) dx$$
$$= p_k \int_{X} \mathbb{1}_{V_k}(x) |\det[Jf_k(x) Jf_k(x)^T]|^{\frac{1}{2}} q(f_k(x)) dx$$
$$= p_k \int_{V_k} |\det[Jf_k(x) Jf_k(x)^T]|^{\frac{1}{2}} q(f_k(x)) dx$$
$$= p_k \int_{\mathcal{Z}} q(z) dz = p_k$$
$$\tag{23}$$

Proviing $(iii)$. Taken together, (22) and (23) yield that $z$ and $k$ are independent random variables since,

$$p(z, k) = \int_{\mathcal{X}} p(x, z, k) dx = p_k q(z) = p(k) p(z) \quad (24)$$

Proving $(iv)$. Dividing (21) by $p(k) = p_k$ we get that the distribution of $x$ conditioned on a particular chart is given by:

$$p(x|k) = \mathbb{1}_{V_k}(x) |\det[Jf_k(x) Jf_k(x)^T]|^{\frac{1}{2}} q(f_k(x)) \quad (25)$$

In particular, $p(x|k)$ is zero unless $x \in U_k$. Meanwhile $p(k|x)$ is given by the Bayes' formula as:

$$p(k|x) = \frac{p(x|k) p(k)}{\sum_{j=1}^{K} p(x|j) p(j)}$$
$$= \frac{p_k \mathbb{1}_{V_k}(x) |\det[Jf_k(x) Jf_k(x)]|^{\frac{1}{2}} q(f_k(x))}{\sum_{j:x \in U_j} p_j |\det[Jf_j(x) Jf_j(x)^T]|^{\frac{1}{2}} q(f_j(x))}$$
$$\tag{26}$$

Proving $(v)$. Note that the distribution $p(k|x)$ is thus also zero unless $x \in U_k$, a fact that will be employed during inference. Finally the density $p(x)$ is given by:

$$p(x) = \sum_{k=1}^{M} p(x|k) p(k)$$
$$= \sum_{k:x \in U_k} p_k |\det[Jf_k(x) Jf_k(x)^T]|^{\frac{1}{2}} q(f_k(x))$$
$$\tag{27}$$

$\square$

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
