# OpenReview forum: "VQ-Flows: Vector Quantized Local Normalizing Flows"
_auai.org/UAI/2022/Conference — UAI 2022 Poster_

### Official Review · Reviewer_Zawj · 2022-04-09

**Q2(1) Originality/Novelty:** 2
**Q2(2) Significance/Impact:** 2
**Q2(3) Correctness/Technical Quality:** 3
**Q2(6) Clarity Of Writing:** 3
**Q6 Overall Score:** 4
**Q8 Confidence In Your Score:** 3

**Q1 Summary And Contributions:**

This paper try to solve the known difficulity for normalizing flow in modeling data with a non-trivial topology. They propose to divide the space and learn local normalizing flow inside eahc divided local space. Specifically, they use VQ-VAE to cluster the data into groups and train the normalizing flow conditioned on the VQ-VAE latent code.

**Q2 Assessment Of The Paper:**

More detailed information regarding each of these aspects is given below:

**Q2(4) Quality Of Experiments (Optional):**

1: Poor: The experimental evaluation is flawed or the results fail to adequately support the main claims.

**Q2(5) Reproducibility:**

3: Good: Key resources (e.g., proofs, code, data) are available and key details (e.g., proofs, experimental setup) are sufficiently well-described for competent researchers to confidently reproduce the main results.

**Q3 Main Strengths:**

1. The proposed method is sound and straightforward. It deals with an interesting and important aspect for normalizing flow based generative model to obey the underlying data geometry.

**Q4 Main Weakness:**

1. The experiments are conducted only on 3D toy datasets, which cannot fully reveal the effectiveness of the model. Actually, for those low-dimnesional data, I believe simpler mode like KDE could already perform well.

**Q5 Detailed Comments To The Authors:**

The paper is well-written.

**Q7 Justification For Your Score:**

lack of experiments: experiments on toy datasets alone will not be of interest to the community.

**Q9 Complying With Reviewing Instructions:**

1: Yes.

---

### Official Review · Reviewer_gC1L · 2022-04-10

**Q2(1) Originality/Novelty:** 2
**Q2(2) Significance/Impact:** 2
**Q2(3) Correctness/Technical Quality:** 3
**Q2(6) Clarity Of Writing:** 3
**Q6 Overall Score:** 4
**Q8 Confidence In Your Score:** 3

**Q1 Summary And Contributions:**

In this paper, the authors aim to overcome some limitations un normalizing flow methods, such as when the data distribution topology is has discontinuities. The contribution relies on conformal embedding flows, recently introduced in [11]. This allows to partition the input space into charts and use what they call “an atlas of charts”, by using a family of normalizing flows conditioned on quantized centers.


**Q2 Assessment Of The Paper:**

More detailed information regarding each of these aspects is given below:

**Q2(4) Quality Of Experiments (Optional):**

3: Good: The experimental evaluation is adequate, and the results convincingly support the main claims.

**Q2(5) Reproducibility:**

3: Good: Key resources (e.g., proofs, code, data) are available and key details (e.g., proofs, experimental setup) are sufficiently well-described for competent researchers to confidently reproduce the main results.

**Q3 Main Strengths:**

The paper addresses an important limitation in normalizing flow methods, which is when dealing with data distributions supported in a low manifold or has a non-trivial topology, such as discontinuities in the data distribution support.

The proposed approach is interesting because it provides a framework that is agnostic to the type of flow transformation employed and retains the key properties of normalizing flows.

**Q4 Main Weakness:**

See “Q5.Detailed Comments To The Authors” for detailed comments with some feedback to enhance the paper and its presentation, including missing references [Gemici et al., 2016] and [Mathieu et al., 2020].


**Q5 Detailed Comments To The Authors:**


The authors need to clearly define what is “an atlas of charts”. This is a key in understanding the paper and its contributions; unfortunately, it is not explained clearly, besides an illustration in Figure 1.

The presentation in the paper can be enhanced by clearly providing definitions and lemma/theorem presentation in order to better structure the main contributions. Moreover, proper definitions should be given in order to help the reader to understand this work, such as the mathematical definition of the locally conformally flat property, which is a main building block of this paper (but its presentation around Eq. (8) is not clear).

A main weakness is that the work seems to entirely rely conformal embedding flows, which was recently proposed by Brendan Leigh Ross and Jesse C Cresswell in [11]. At the end, it seems that the proposed method operates k conformal embedding flows, which allows to define what is called charts. In this sense, the contribution of this work is not major.

There has been many studies for normalizing flows on manifolds. The authors did not cite them, and neither compare with these methods. Some references are:
- M. C. Gemici, D. Rezende, and S. Mohamed. Normalizing Flows on Riemannian Manifolds, ArXiv, abs/1611.02304, 2016.
- E. Mathieu and M. Nickel. Riemannian Continuous Normalizing Flows. In Advances in Neural Information Processing Systems, volume 33, pages 2503–2515, 2020.

It is not clear the effect of the number of partitions “k” on the performance. While there is a small section on ablation study to analyze it, it is not well presented and opens the way to new questions, such as the relation between k, the number of partitions, and k in the k-means.

There are some spelling and grammatical errors, such as “paramaterizes”, “generated sampes”


**Q7 Justification For Your Score:**

Our overall assessment is based on all the aforementioned comments. The main weaknesses being that the contribution seems not to be major, and some missing comparative analysis with well-know methods for normalizing flows on manifolds, such as [Gemici et al., 2016] and [Mathieu et al., 2020].

**Q9 Complying With Reviewing Instructions:**

1: Yes.

---

### Official Review · Reviewer_p1EA · 2022-04-11

**Q2(1) Originality/Novelty:** 3
**Q2(2) Significance/Impact:** 3
**Q2(3) Correctness/Technical Quality:** 3
**Q2(6) Clarity Of Writing:** 3
**Q6 Overall Score:** 6
**Q8 Confidence In Your Score:** 3

**Q1 Summary And Contributions:**

The paper proposes a mixture of local flows by utilizing the idea of vector quantized autoencoders. The proposed idea is presented on many 3D problems with various topologies on which flow-based models like RealNVP or MAF.

**Q2 Assessment Of The Paper:**

More detailed information regarding each of these aspects is given below:

**Q2(4) Quality Of Experiments (Optional):**

2: Fair: The experimental evaluation is weak: important baselines are missing, or the results do not adequately support the main claims.

**Q2(5) Reproducibility:**

2: Fair: Key resources (e.g., proofs, code, data) are unavailable but key details (e.g., proof sketches, experimental setup) are sufficiently well-described for an expert to confidently reproduce the main results.

**Q3 Main Strengths:**

+ The idea is neat and makes sense.
+ The discussion of related work is in-depth and valuable for understanding the proposed approach.
+ The experiments are insightful.

**Q4 Main Weakness:**

- It would be beneficial to check how the proposed approach works on more powerful flow-based generative models: (a) Residual flows: Chen, R. T., Behrmann, J., Duvenaud, D. K., & Jacobsen, J. H. (2019). Residual flows for invertible generative modeling. Advances in Neural Information Processing Systems, 32. (b) Invertible DenseNets: Perugachi-Diaz, Y., Tomczak, J., & Bhulai, S. (2021). Invertible densenets with concatenated lipswish. Advances in Neural Information Processing Systems, 34.

- The experiments nicely present the idea and current deficiencies of flow-based deep generative models. However, it would be beneficial to see how the VQ-flows work on high-dim problems (e.g., images).

**Q5 Detailed Comments To The Authors:**

Please see Q4.

**Q7 Justification For Your Score:**

The paper is well-written and easy to follow. The idea is interesting and has great potential. The experiments present the core idea, however, they are limited to 3D problems. It would be interesting to see how the proposed idea works on other flow-based models and higher-dim problems (e.g., images).

**Q9 Complying With Reviewing Instructions:**

1: Yes.

---

### Official Review · Reviewer_sieb · 2022-04-12

**Q2(1) Originality/Novelty:** 3
**Q2(2) Significance/Impact:** 3
**Q2(3) Correctness/Technical Quality:** 4
**Q2(6) Clarity Of Writing:** 4
**Q6 Overall Score:** 8
**Q8 Confidence In Your Score:** 3

**Q1 Summary And Contributions:**

A new normalising flow is proposed to improve modelling of lower-dimensional manifolds and more complex topologies, by constructing a composition of multiple local flows using a vector quantization. Clear artificial data experiments are used to demonstrate improved modelling capability in density estimation and sample generation tasks, but more complex tasks are not considered.

**Q2 Assessment Of The Paper:**

More detailed information regarding each of these aspects is given below:

**Q2(4) Quality Of Experiments (Optional):**

3: Good: The experimental evaluation is adequate, and the results convincingly support the main claims.

**Q2(5) Reproducibility:**

3: Good: Key resources (e.g., proofs, code, data) are available and key details (e.g., proofs, experimental setup) are sufficiently well-described for competent researchers to confidently reproduce the main results.

**Q3 Main Strengths:**

Extremely well written paper that manages to present a fairly complex model so that it is easy to understand. Clear and intuitive contribution to the normalising flow literature that still required non-trivial technical development.

**Q4 Main Weakness:**

Empirical experiments on some more real/complex scenarios would have been nice.

**Q5 Detailed Comments To The Authors:**

This is overall a good work and I do not have any major issues. The paper provides one of the best descriptions of normalizing flows I have read and it is easy to understand how the proposed method works and why it is needed -- I could well use this paper on a course as an example of a paper introducing difficult technical ideas in a readable manner.

Even though combining multiple local flows has been done before (using mixtures), the reasoning of why VQ is to be preferred is sound and the technical details of the solution seem reasonable. The idea is not the most novel and innovative one, but some insight was required.

The artificial data experiments are clear and especially Figure 4 is telling a very clear story, but the experiments could be a bit broader. Some sort of comparison on real-data benchmarks would have been useful to highlight how well the improvement translates to more complex modelling problems. I think the work is valuable even if the current solution did not yet work on some of the more complex scenarios or it some other recent advances also solve the same problems, but it would be useful for the reader to get better insight on what to expect. Also, a (visual) comparison against mixtures of flows [31-33] would make the paper stronger as they would intuitively be the closest alternatives.

Minor remarks: Fix capitalisation in references (e.g. [31] should be "neural ODEs"). Page 3 has two terribly long paragraphs (one in related work, on around Eq. (5)) that should be split into two or even three paragraphs.

**Q7 Justification For Your Score:**

Clear contribution for the normalizing flow literature, introducing well motivated new ideas in a format that is easy to digest and demonstrating good empirical performance (though only in artificial examples).

**Q9 Complying With Reviewing Instructions:**

1: Yes.

---

### Decision · Program_Chairs · 2022-05-15

**Decision:**

Accept (Poster)

**Comment:**

Meta Review: The paper introduces a new class of models for learning low-dimensional distributions, while still retaining normalizing flows as a building block. (The latter have shown great promise, but are designed to work for full-dimensional distributions.) The main idea is to "stitch" multiple local charts for the manifold, each locally parametrized by a normalizing flow. The reviewers and myself agree that the idea behind the model introduced is very nice and promising and the paper is well-written (despite the relatively heavy amount of math required). The experimental evaluation has some flaws, but they don't outweigh the conceptual novelty in the paper.